# Analyzing Norm Violations in Live-Stream Chat

**Jihyung Moon**[1*], **Dong-Ho Lee**[1,2*], **Hyundong Cho**[2], **Woojeong Jin**[2], **Chan Young Park**[3],
**Minwoo Kim**[4], **Jonathan May**[2], **Jay Pujara**[2], **Sungjoon Park**[1]

[1]SoftlyAI Research, [2]University of Southern California,
[3]Carnegie Mellon University, [4]Selectstar

{jihyung.moon,dongho.lee,sungjoon.park}@softly.ai {jcho,jonmay,jpujara}@isi.edu

{dongho.lee,woojeong.jin}@usc.edu {chanyoun}@cs.cmu.edu {mwkim}@selectstar.ai

## Abstract

Toxic language, such as hate speech, can deter users from participating in online communities and enjoying popular platforms. Previous approaches to detecting toxic language and norm violations have been primarily concerned with conversations from online forums and social media, such as Reddit and Twitter. These approaches are less effective when applied to conversations on live-streaming platforms, such as Twitch and YouTube Live, as each comment is only visible for a limited time and lacks a thread structure that establishes its relationship with other comments. In this work, we share the first NLP study dedicated to detecting norm violations in conversations on live-streaming platforms. We define norm violation categories in live-stream chats and annotate 4,583 moderated comments from Twitch. We articulate several facets of live-stream data that differ from other forums, and demonstrate that existing models perform poorly in this setting. By conducting a user study, we identify the informational context humans use in live-stream moderation, and train models leveraging context to identify norm violations. Our results show that appropriate contextual information can boost moderation performance by 35%. [1]

## 1 Introduction

Interactive live streaming services such as Twitch [2] and YouTube Live [3] have emerged as one of the most popular and widely-used social platforms. Unfortunately, streamers on these platforms struggle with an increasing volume of toxic comments and norm-violating behavior.[4] While there has been extensive research on mitigating similar problems for online conversations across various platforms such as Twitter ([Waseem and Hovy, 2016](); [Davidson]()

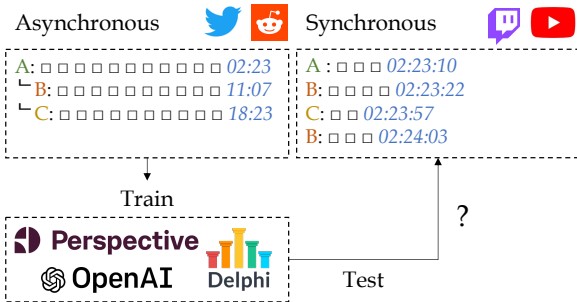

Figure 1: **A Motivating Example.** Chat in the synchronous domain has different characteristics than those in the asynchronous domain: (1) the temporal gap between chats and message length are much smaller; and (2) relationships between chats are less clearly defined. Such differences make chats in the synchronous domain more difficult to be moderated by existing approaches.

[et al., 2017](); [Founta et al., 2018](); [Basile et al., 2019](); [ElSherief et al., 2021]()), Reddit ([Datta and Adar, 2019](); [Kumar et al., 2018](); [Park et al., 2021]()), Stackoverflow ([Cheriyan et al., 2017]()) and Github ([Miller et al., 2022]()), efforts that extend them to live streaming platforms have been absent. In this paper, we study unique characteristics of comments in live-streaming services and develop new datasets and models for appropriately using contextual information to automatically moderate toxic content and norm violations.

Conversations in online communities studied in previous work are *asynchronous*: utterances are grouped into threads that structurally establish conversational context, allowing users to respond to prior utterances without time constraints. The lack of time constraints allows users to formulate longer and better thought-out responses and more easily reference prior context.

On the other hand, conversations on live streaming platforms are *synchronous*, i.e. in real-time, as utterances are presented in temporal order without a thread-like structure. Context is mostly established by consecutive utterances ([Li et al., 2021]()). The transient nature of live-stream utterances en-

---

*Authors contributed equally.

[1]https://github.com/softly-ai/live-NormVio

[2]https://www.twitch.tv/

[3]https://www.youtube.com/

[4]https://safety.twitch.tv/s/article/Community-Guidelines

courages fast responses, and encourages producing multiple short comments that may be more prone to typos (70% of comments are made up of $< 4$ words). Figure 1 shows an illustration of the contrasting temporal and length patterns between the asynchronous and synchronous platforms.

Owing to these different characteristics, we find that previous approaches for detecting norm violations are ineffective for live-streaming platforms. To address this limitation, we present the first NLP study of detecting norm violations in live-stream chats. We first establish norms of interest by collecting 329 rules from Twitch streamers' channels and define 15 different fine-grained norm categories through an iterative coding process. Next, we collect 4,583 moderated chats and their corresponding context from Twitch live streams and annotate them with these norm categories (§2.1-§2.3). With our data, we explore the following research questions: (1) How are norm violations in live-stream chats, *i.e.* synchronous conversations, different from those in previous social media datasets, *i.e.* asynchronous conversations?; (2) Are existing norm violation or toxicity detection models robust to the distributional shift between the asynchronous and synchronous platforms? (§3.1, §3.3); and (3) Which features (*e.g.,* context and domain knowledge) are important for detecting norm violation in synchronous conversations? (§3.2)

From our explorations, we discover that (1) live-stream chats have unique characteristics and norm violating behavior that diverges from those in previous toxicity and norm-violation literature; (2) existing models for moderation perform poorly on detecting norm violations in live-stream chats; and (3) additional information, such as chat and video context, are crucial features for identifying norm violations in live-stream chats. We show that incorporating such information increases inter-annotator agreement for categorizing moderated content and that selecting temporally proximal chat context is crucial for enhancing the performance of norm violation detection models in live-stream chats.

## 2 NormVio-RT

To investigate norm-violations in live-stream chat, we first collect Norm Violations in Real-Time Conversations (**NormVio-RT**), which contains 4,583 norm-violating comments on Twitch that were moderated by channel moderators.[5] An overview of our

[5]Please contact the authors for the anonymized study data.

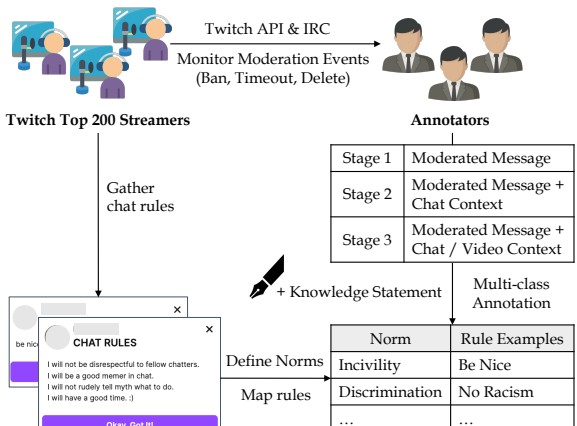

Figure 2: **Data Construction**. Norms are manually defined based on the chat rules of the top 200 streamers, and annotators annotate the violated norm of moderated event by three stages.

data collection procedure is illustrated in Figure 2. We first select 200 top Twitch streamers and collect moderated comments from their streamed sessions (§2.1). To understand why these chats are moderated, we collect chat rules from these streamers and aggregate them to define coarse and fine-grained norm categories (§2.2). We design a three-step annotation process to determine the impact of the chat history, video context, and external knowledge on labeling decisions (§2.3). Lastly, we present analysis of the collected data (§2.4).

### 2.1 Data Collection

We collected data using the Twitch API and IRC[6] from the streamers with videos that are available for download among the top 200 Twitch streamers as of June 2022[7]. We specifically looked for comments that triggered a moderation event during a live stream (e.g. *user ban*, *user timeout*), and collected the moderated comment and the corresponding video and chat logs up to two minutes prior to the moderation event. Logs of moderated events from August 22, 2022 to September 3, 2022 were collected. We excluded comments that were moderated within less than 1 second of being posted, as they are likely to have been moderated by bots rather than humans.

### 2.2 Norm Categorization

Twitch streamers can set their own rules for their channels, and these channel-specific rules are essential for understanding why comments were moderated. We first collect 329 rules from the top

[6]https://github.com/TwitchIO/TwitchIO
[7]https://twitchtracker.com/channels/viewership/english

| Coarse | Fine-grained | Target | Rule Examples |
|---|---|---|---|
| Discrimination | Discrimination | - | No racism, sexism or homophobia. |
| HIB (Harassment, Intimidation, Bullying) | HIB | Broadcaster OIB OOB | No HIB towards broadcaster
No HIB towards viewers, moderators, etc.
No HIB towards other broadcasters, politicians, etc. |
| Privacy | Doxing | - | Please no personal questions about me.
Do not share any personal information about yourself or others. |
| Inappropriate Contents | NSFW
Self-destructive
Illegal
Spoiler | -
-
-
- | No NSFW content (e.g., Inappropriate ASCII arts).
No talk of suicide.
No drug discussion of any kind.
Do not give game spoilers. |
| Off Topic | Controversial Topic
Begging | -
- | No drama, politics or religion.
No begging for subscriptions or money. |
| Spam | Excessive & Repetitive
Advertisements | -
- | No walls of text.
No self promotion unless authorized. |
| Meta-Rules (Live streaming specific) | Backseating & Tall order
Mentioning other broadcasters
Specific language only | -
-
- | Don't tell me what to do. Don't ask for mod.
Don't talk down on other streamers.
English only. |
| Incivility (Miscellaneous) | Incivility | - | Be nice, Be civil |

Table 1: **Live streaming norms.** We map rules from top 200 Twitch streamers' channels to coarse and fine-grained level norms. Some rules specify targets (**OOB**: **O**thers **O**utside of **B**roadcast, **OIB**: **O**thers **I**n **B**roadcast).

200 Twitch streamers' channels. Next, following Fiesler et al. (2018), we take an iterative coding process such that the authors of this paper individually code for rule types with certain categories, come together to determine differences and then repeat the coding process individually. With this process, we aggregated similar rules into 15 different fine-grained level norm categories (*e.g.*, controversial topics, begging) and cluster multiple fine-grained categories into 8 different coarse-grained norm categories (e.g., off-topic). To better understand the *targets* of offensive comments in the HIB (Harassment, Intimidation, Bullying) class, we added an additional dimension to consider whether the target is the broadcaster (streamer), participants in the channel (e.g., moderators and viewers), or someone not directly involved in the broadcast. We asked annotators to assign "Incivility" to cases where annotators do not believe that a specific pre-defined rule type has been violated although moderated. Examples of "Incivility" are provided in Appendix A.4. Table 1 shows the resulting norm categories and corresponding fine-grained norms with examples.

## 2.3 Violated Norm Type Annotation

We recruited three annotators who are fluent in English and spend at least 10 hours a week on live streaming platforms to ensure that annotators understood live streaming content and conventions. Their fluency was verified through several rounds of pilot annotation work. Internal auditors continuously conducted intermittent audits to ensure that annotators fully understood the guidelines.

Annotators were asked to annotate each mod-

| Knowledge | Template |
|---|---|
| Platform | ** is {emoji, text} that means *<explanation>* |
| Streamer | *<streamer> <explanation>* |

Table 2: **A knowledge statement template.**

eration event (i.e. moderated comment) with the rule types it violates. To measure the importance of context in determining types of norm violations, annotators were asked to provide labels for three different scenarios with varying amounts of context: (1) **Stage 1**: annotate based on only the user's last message before the moderation event (*single utterance*); (2) **Stage 2**: annotate based on chat logs up to two minutes prior to the moderation (*+chat context*); (3) **Stage 3**: annotate based on chat logs and their corresponding video clip of the same duration (*+video context*). Since rules are not mutually exclusive (*e.g.,* a message can violate both discrimination & harassment), they are allowed to choose multiple rule types if there are more than one violated rule at each stage. All the annotations are done with our internal annotation user interface (See Appendix A.1). To determine the final label for each moderated event, we aggregate the labels of annotators using a *majority vote* with heuristic rules (See Appendix A.2).

Lastly, to examine how much external knowledge matters in understanding comments on live streaming platforms, we asked annotators to (1) indicate whether external knowledge is necessary to understand why a comment triggered a moderation event and if so (2) describe what that knowledge is. We focus on two types of external knowledge: platform- and streamer-specific. Platform-specific knowledge includes the implicit meaning of partic-

| Coarse | Fine-grained | # Rules | # Violates | | |
|---|---|---|---|---|---|
| | | | stage 1 | stage 2 | stage 3 |
| Discrimination | Discrimination | 13.98% (46) | 2.34% (104) | 2.25% (101) | 2.34% (105) |
| HIB | HIB | 22.49% (74) | 21.33% (947) | 26.55% (1,190) | 27.80% (1,246) |
| Privacy | Doxing | 0.60% (2) | 0.34% (15) | 0.36% (16) | 0.36% (16) |
| Inappropriate Contents | Spoiler | 0.60% (2) | 0.02% (1) | 0.02% (1) | 0.02% (1) |
| | NSFW | 1.82% (6) | 0.86% (38) | 0.85% (38) | 0.85% (38) |
| | Self-destructive | 1.21% (4) | 0.32% (14) | 0.29% (13) | 0.29% (13) |
| | Illegal | 0.30% (1) | 0.16% (7) | 0.07% (3) | 0.07% (3) |
| Off Topic | Controversial Topic | 5.47% (18) | 0.59% (26) | 0.85% (38) | 0.83% (37) |
| | Begging | 1.51% (5) | 1.44% (64) | 1.36% (61) | 1.36% (61) |
| Spam | Excessive & Repetitive | 11.24% (37) | 17.59% (781) | 21.64% (970) | 21.42% (960) |
| | Advertisements | 11.24% (37) | 4.64% (206) | 4.40% (197) | 4.42% (198) |
| Meta-Rules (Live streaming specific) | Mentioning other streamers | 14.28% (47) | 0.72% (32) | 10.62% (476) | 10.58% (474) |
| | Backseating & Tall order | 5.16% (17) | 3.45% (153) | 3.70% (166) | 3.77% (169) |
| | Specific language only | 10.03% (33) | 0.97% (43) | 6.94% (311) | 6.94% (311) |
| Incivility (Miscellaneous) | Incivility | - | 12.30% (546) | 11.57% (519) | 11.51% (516) |
| | Non-Identifiable | - | 32.93% (1,462) | 8.52% (382) | 7.45% (334) |
| Total | | 329 | 4,439 | 4,482 | 4,482 |

Table 3: **Data Statistics.** *# of rules* indicates the number of streamers specifying the norm in their channels and *# of violates* indicates actual number of messages that violate corresponding norms.

ular emojis, emotes, and slang that are commonly used on Twitch. Streamer-specific knowledge involves the streamer's personal background and previous streaming sessions. As shown in Table 2, we provide templates for each type that annotators can easily fill out (More details in Appendix A.3).

## 2.4 Data Statistics and Analysis

**General Observations** We identified three characteristics that distinguish real-time live-streaming chat from other domains. First, the majority of comments are very short; 70% of comments are made up of < 4 words. Additionally, they are often very noisy due to the real-time nature of communication, which leads to a high number of typos, abbreviations, acronyms, and slang in the comments. Lastly, some comments use unusual visual devices such as ASCII art and "all caps", to make them more noticeable. This is because each comment is visible only for a short time in popular streams (on average, there are around 316 chats per minute for the streamers in our data). The chat window in live streaming platforms can only display a limited number of comments, so viewers are incentivized to use visual devices to draw the streamer's attention in these fast-paced conditions.

**False positives in data.** We find that the "Incivility" case contains many false positives, as they include cases that seem to have been moderated for no particular reason. We asked annotators to put all miscellaneous things into the "Incivility" category, and also to mark as "Incivility" if they

| % Agreement | Stage 1 | Stage 2 | Stage 3 |
|---|---|---|---|
| Exact Match | 39.71% (1,820) | 41.10% (1,884) | 40.67% (1,864) |
| Partial Match | 54.11% (2,480) | 75.03% (3,439) | 75.21% (3,447) |
| Majority Vote | 96.85% (4,439) | 97.79% (4,482) | 97.79% (4,482) |

Table 4: **Inter-annotator Agreement.** Presents the percentage of moderated events for 4,583 events while the number in parentheses indicates the number of events.

could not identify any reason for the moderation. We found that many cases are not identifiable, as shown in Table 3. It is natural that many cases are non-identifiable in stage 1, as annotators are only given the moderated comment and no context. However, the 7.45% non-identifiable cases that remain even after stage 3 could be false positives, or they could be cases where the moderation event occurred more than two minutes after a problematic comment was made.

**Context improves inter-annotator agreement.** Interestingly, providing context helps mitigate annotator bias, as shown by the increase in inter-annotator agreement from stage 1 to stages 2 and 3 in Table 4. Here, the *exact match* determines whether all three annotators have exactly the same rules; *partial match* determines whether there is at least one intersection rule between three annotators; and *majority vote* chooses the rule types that were selected by at least two people. Also, non-identifiable and disagreement cases drop significantly when the contexts are given as shown in Table 3. Similarly for determining rule types, context also helps annotators identify targets for HIB and reduces inconsistencies between annota-

tors. Our observation emphasizes the importance of context in synchronous communication and differs from previous findings that context-sensitive toxic content is rare in asynchronous communication (Pavlopoulos et al., 2020; Xenos et al., 2021). Analysis details are in Appendix A.2.

**External knowledge helps annotations.** To investigate the impact of external knowledge on annotators' labeling decisions, we compare annotations made with and without external knowledge provided. For examples with knowledge statements, we expect to see differences in annotation if external knowledge is necessary to comprehend why they were moderated. Statistics show that 296 examples (6.6%) require knowledge, with 183 examples requiring streamer knowledge and 187 examples requiring platform knowledge. Note that there are some examples require both. Details of statistics and examples are presented in Appendix A.3.

**Norm Category Distribution** Table 3 shows the norm category distribution of streamers' rules and the moderated comments. While the categories are not directly comparable to the ones defined in NormVio for Reddit (Park et al., 2021), we identified a few similar patterns. First, in both domains, Harassment and Incivility (i.e., Discrimination, HIB, Incivility) take up a significant portion of the entire set of norm violations. Also, the two domains show a similar pattern where rules for Off-Topic, Inappropriate Contents, and Privacy exist but are relatively less enforced in practice. However, we also found that the two domains differ in various ways. For example, Spam and Meta-Rules cover significantly higher portions of both rules and moderated comments on Twitch than on Reddit. On the other hand, there are fewer rules about content on Twitch, which implies that streamers are less concerned about the content of the comments than Reddit community moderators. As our data shows that norm-violating comments on live chats exhibit distinctive rules and patterns, it suggests that the existing norm violation detection systems may not perform well without domain adaptation to account for these distributional differences. We examine this hypothesis empirically in the following section and suggest appropriate modeling adjustments to better detect toxicity for real-time comments.

## 3 Toxicity Detection in Live-stream Chat

In this section, we first check whether norm violation and toxicity detection models are robust

| Model | Precision | Recall | F1 |
|---|---|---|---|
| ToxiGen | 0.31 | 0.91 | 0.46 |
| Perspective API | 0.39 | 0.95 | 0.56 |
| OpenAI moderation | 0.11 | 0.94 | 0.20 |
| OpenAI content filter | 0.55 | 0.86 | 0.67 |

Table 5: **Performance (Binary F1) of toxicity detection models** on HIB and Discrimination data. Binary F1 refers to the results for the 'toxic' class.

to the distributional shift from asynchronous conversations to synchronous conversations and vice versa, and identify how important the context or domain knowledge are for detecting toxicity and norm violation in synchronous conversations.

### 3.1 Performance of Existing Frameworks.

To examine the difference in toxicity detection between asynchronous and synchronous communication, we investigate whether existing toxicity detection models are effective for synchronous communication. We evaluate the performance of four existing tools on NormVio-RT: Google's Perspective API (Lees et al., 2022)[8], OpenAI content filter[9], OpenAI moderation (Markov et al., 2022)[10], and a RoBERTa-large model fine-tuned on machine-generated toxicity dataset called ToxiGen (Hartvigsen et al., 2022). We only use examples from the discrimination and HIB categories in NormVio-RT, as they are most similar to the label space that the existing models are trained for (*e.g.,* hateful content, sexual content, violence, self-harm, and harassment). Categories are determined based on the stage 1 consolidated labels, as we do not provide any context to the model. Additionally, we select an equal number of random chats from the collected stream to construct negative examples. To ensure the quality of negative examples, we only select chats that are not within two minutes prior to any moderation event as they are less likely to contain norm violating chats. We also only select chats from users who have never been moderated in our data. To obtain the predictions from the models, we check whether toxicity score is greater than or equal to 0.5 for Perspective API, and for OpenAI, check the value of the "flagged" field which indicates whether OpenAI's content policy is violated. We use binary classification outputs for ToxiGen.

Table 5 shows the results obtained from 2,102 examples with 1,051 examples each for toxic and non-

---

[8]https://perspectiveapi.com/
[9]https://beta.openai.com/docs/models/content-filter
[10]https://beta.openai.com/docs/api-reference/moderations

| Context | All | Discrimination | HIB | Privacy | Inapt. Contents | Off Topic | Spam | Meta-Rules | Incivility |
|---|---|---|---|---|---|---|---|---|---|
| - | 0.70 $_{\pm 0.00}$ | 0.11 $_{\pm 0.00}$ | 0.52 $_{\pm 0.01}$ | **0.05** $_{\pm 0.03}$ | 0.12 $_{\pm 0.01}$ | 0.07 $_{\pm 0.00}$ | 0.63 $_{\pm 0.01}$ | **0.65** $_{\pm 0.01}$ | 0.28 $_{\pm 0.04}$ |
| Single-user context | 0.78 $_{\pm 0.00}$ | 0.07 $_{\pm 0.02}$ | 0.50 $_{\pm 0.01}$ | 0.03 $_{\pm 0.02}$ | **0.18** $_{\pm 0.05}$ | 0.05 $_{\pm 0.00}$ | 0.67 $_{\pm 0.02}$ | 0.58 $_{\pm 0.04}$ | 0.28 $_{\pm 0.06}$ |
| Multi-user context (event) | 0.75 $_{\pm 0.04}$ | 0.03 $_{\pm 0.00}$ | 0.44 $_{\pm 0.02}$ | 0.01 $_{\pm 0.00}$ | 0.14 $_{\pm 0.12}$ | 0.05 $_{\pm 0.01}$ | 0.66 $_{\pm 0.00}$ | 0.60 $_{\pm 0.03}$ | 0.17 $_{\pm 0.00}$ |
| Multi-user context (utterance) | 0.91 $_{\pm 0.05}$ | 0.04 $_{\pm 0.00}$ | 0.61 $_{\pm 0.05}$ | 0.00 $_{\pm 0.00}$ | 0.09 $_{\pm 0.03}$ | 0.10 $_{\pm 0.04}$ | 0.66 $_{\pm 0.01}$ | **0.65** $_{\pm 0.04}$ | 0.24 $_{\pm 0.12}$ |
| Multi-user context (first) | **0.95** $_{\pm 0.00}$ | 0.05 $_{\pm 0.01}$ | **0.61** $_{\pm 0.01}$ | 0.01 $_{\pm 0.00}$ | 0.11 $_{\pm 0.02}$ | 0.08 $_{\pm 0.03}$ | **0.70** $_{\pm 0.03}$ | 0.62 $_{\pm 0.02}$ | **0.45** $_{\pm 0.03}$ |
| Broadcast category | 0.77 $_{\pm 0.03}$ | **0.13** $_{\pm 0.03}$ | 0.48 $_{\pm 0.01}$ | 0.02 $_{\pm 0.01}$ | 0.13 $_{\pm 0.04}$ | 0.13 $_{\pm 0.05}$ | 0.65 $_{\pm 0.01}$ | 0.64 $_{\pm 0.02}$ | 0.30 $_{\pm 0.02}$ |
| Rule text | 0.75 $_{\pm 0.01}$ | 0.05 $_{\pm 0.08}$ | 0.11 $_{\pm 0.17}$ | 0.00 $_{\pm 0.00}$ | 0.12 $_{\pm 0.06}$ | **0.29** $_{\pm 0.18}$ | 0.58 $_{\pm 0.04}$ | 0.38 $_{\pm 0.02}$ | 0.13 $_{\pm 0.03}$ |

Table 6: **Performance on Norm Classification.** macro F1 score for each coarse-level norm category. "All" refers to binary classification between moderated and unmoderated messages without considering norm category. Best models are **bold** and second best ones are underlined. Scores are average of 3 runs (3 random seeds).

toxic messages. The results illustrate that while existing models do not frequently produce false positives (high recall), they perform poorly in detecting toxic messages found in synchronous chats, with a detection rate of only around 55% at best (low precision).

## 3.2 Norm Classification in NormVio-RT.

To understand the model's ability to detect norm violations and how additional information can affect detection, we train binary classification models for each category with different types of context including conversation history, broadcast category, and rule description following Park et al. (2021).

**Experimental Setup.** For each coarse-level category, we train a RoBERTa-base model with a binary cross entropy loss to determine whether the message is violating the certain norm or not. Following Park et al. (2021), we perform an 80-10-10 train/dev/test random split of moderated messages and add the same number of unmoderated messages in the same split. Next, for each binary classification, we consider the target category label as 1 and others as 0 and construct a balanced training data set. Appendix B (See Table 12). Here, the labels are based on stage 3.

To examine how context affects model performance, we experiment with four model variants with different input context: (1) **Single user context** is only the chat logs of the moderated user that took place up to two minutes before the moderation event; (2) **Multi-user context (event)** is $N$ messages that directly precede the moderation event, regardless of whether it belongs to the moderated user; (3) **Multi-user context (utterance)** is $N$ messages that directly precedes the *single utterance*, which is the moderated user's last message before the moderation event (*i.e.,* chat 3 in Figure 3).; (4) **Multi-user context (first)** is the first $N$ messages of the collected two-minute chat logs. The intuition for this selection is that the moderation event may have taken place much earlier than

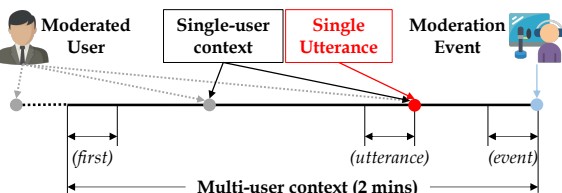

Figure 3: **Multi-user context** is chat logs that occurred up to two minutes before the moderation event while **single-user context** is chat logs of moderated user in a multi-user context and **single utterance** is the moderated user's last message before the moderation event.

the moderation event. In all the Multi-user contexts, we use $N = 5$; (5) **Broadcast category** is the category that streamers have chosen for their broadcast. It usually is the title of a game or set to "just chatting"; and (6) **Rule text** is a representative rule example shown in Table 1. The rule text is only used for training examples because it is not possible to know which rule was violated for unseen examples and we use randomly selected rule text for unmoderated negative examples in training examples. All contexts are appended to the input text (*single utterance*) with a special token ([SEP]) added between the input text and the context. Chat logs for multi-user context and single-user context are placed sequentially with spaces between chats. Training details and data statistics are presented in Appendix B.

**Experimental Results.** Table 6 presents performance of norm classification for coarse-level norm categories. "All" refers to binary moderation detection, whether the message is moderated or not, and not the specific norm type. First, we can see that additional context improves the performance of "All," but context does not consistently improve the performance of category-specific norm classifiers. For example, context reduces performance for categories where the issues are usually limited to the utterance itself (*e.g.,* discrimination and privacy). In contrast, categories that rely on the relationships between utterances, such as HIB and incivility, show improved performance with con-

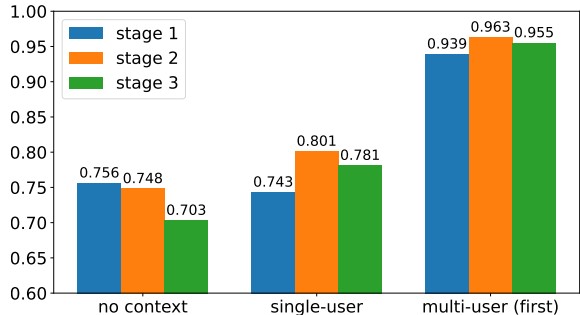

Figure 4: **Performance (F1 Score) of moderation detection** by different ground truth label for each context.

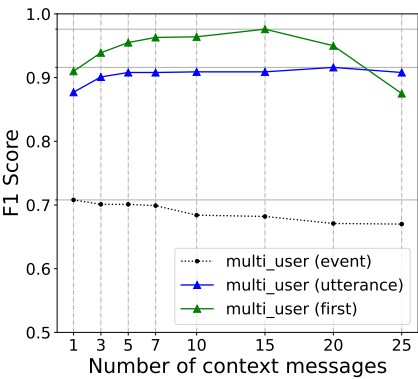

Figure 5: **Performance (F1 Score) trend** of moderation detection with varying context length.

text. Secondly, multi-user context performs quite well compared to the other contexts, indicating that a more global context that includes utterances from other users helps determine the toxicity of target utterances. Lastly, the strong performance of Multi-user context (first) suggests that earlier messages in the two-minute window are more important, meaning that the temporal distance between the moderation event and the actual offending utterance may be substantial in many cases. Thus, our results encourage future efforts on developing a more sophisticated approach for context selection.

**Availability of Context.** To compare human decisions with those of our models, we conduct experiments varying the context available to annotators and models. For example, we expect models trained with only single utterances to perform best when using stage 1 (utterance only) labels as ground-truth labels since humans are also not given any context at stage 1. Indeed, in Figure 4, using the stage 1 labels as the ground truth labels yields the best performance for a model trained without any context, while using the stage 2 (context) labels as the ground truth labels shows the best performance for a model trained with previous chat history. Since our experiments only handle text inputs, it is not surprising that using stage 3 (video) labels as ground-truth labels yields worse performance than using stage 2 labels. However, interestingly, the gap is not large, which indicates that gains from a multi-modal model that incorporates information from the video may be small and that single modality (text-only) models can be sufficient for the majority of moderation instances.

**Context Size.** To understand how the amount of available context affects moderation performance, we compare the multi-user context configurations with various number of messages from one to 25. Figure 5 demonstrates that 15 to 20 messages prior

to the moderated user's message helps with moderation performance the most (See *utterance* and *first*). However, increasing the number of messages that directly precede the moderation event actually lowers moderation performance (See *event*). It may be that most of this context serves as noise.

### 3.3 Distribution Shift in Norm Classification.

Existing tools often focus on identifying harmful speech, but NormVio (Park et al., 2021) also considers a wider range of norm-violating comments on Reddit, similar to NormVio-RT but in a different domain. We compare NormVio and NormVio-RT by evaluating the performance of a model fine-tuned on NormVio with NormVio-RT, and vice versa, to examine the impact of distribution shift between these domains. We choose six coarse-level categories that overlap between the two, as shown in Table 7. To measure with-context performance, we use the previous comment history for Reddit and multi-user context (utterance) for Twitch to simulate the most similar setup in both domains. Overall, experimental results show a pronounced distribution shift between Reddit (*asynchronous*) and Twitch (*synchronous*). Interestingly, models trained on Twitch are able to generalize better than models trained on Reddit despite having 6x less training data. Specifically, models trained using the out-of-domain Twitch+context data perform comparably on the Reddit test set to those trained using in-domain Reddit+context data.

## 4 Related Work

**Toxicity Detection** Most toxic language data consists of explicit hate speech consisting of hate lexicons (Waseem and Hovy, 2016; Davidson et al., 2017; Founta et al., 2018; Basile et al., 2019), group identifiers (Warner and Hirschberg, 2012; Kennedy et al., 2020), and hateful phrase (Silva

| Category | | Without Context | | | | With Context | | | |
|---|---|---|---|---|---|---|---|---|---|
| Reddit (Normvio) | Twitch (Normvio-RT) | R | T→R | T | R→T | R | T→R | T | R→T |
| ALL | ALL | 0.99 ±0.00 | 0.84 ±0.04 | 0.70 ±0.00 | 0.67 ±0.00 | 0.99 ±0.00 | 0.98 ±0.00 | 0.91 ±0.01 | 0.67 ±0.00 |
| Incivility | Incivility | 0.67 ±0.00 | 0.16 ±0.09 | 0.28 ±0.04 | 0.09 ±0.03 | 0.74 ±0.00 | 0.56 ±0.14 | 0.24 ±0.12 | 0.09 ±0.03 |
| Harassment | HIB, Privacy | 0.34 ±0.01 | 0.19 ±0.01 | 0.51 ±0.01 | 0.27 ±0.01 | 0.41 ±0.00 | 0.20 ±0.00 | 0.62 ±0.02 | 0.26 ±0.03 |
| Spam | Spam | 0.47 ±0.02 | 0.22 ±0.02 | 0.63 ±0.01 | 0.28 ±0.01 | 0.53 ±0.01 | 0.27 ±0.01 | 0.66 ±0.01 | 0.28 ±0.01 |
| Off Topic | Off Topic | 0.25 ±0.02 | 0.12 ±0.01 | 0.07 ±0.00 | 0.00 ±0.00 | 0.28 ±0.00 | 0.12 ±0.02 | 0.10 ±0.04 | 0.00 ±0.00 |
| Hate Speech | Discrimination | 0.17 ±0.02 | 0.05 ±0.04 | 0.11 ±0.00 | 0.02 ±0.00 | 0.19 ±0.00 | 0.06 ±0.04 | 0.04 ±0.00 | 0.02 ±0.00 |
| Content | Inapt. Contents | 0.30 ±0.06 | 0.08 ±0.03 | 0.12 ±0.01 | 0.00 ±0.00 | 0.37 ±0.01 | 0.05 ±0.02 | 0.09 ±0.03 | 0.00 ±0.00 |

Table 7: **Performance on distribution shift between norm violations in Reddit and Twitch.** Macro F1 scores for each overlapped norm category. Scores are average of 3 runs (3 random seeds).

et al., 2016) in asynchronous communication (e.g., Twitter, Reddit). However, models trained on such data may have spurious correlations that result in many false positives (e.g., group identifiers) (Sap et al., 2019; Kennedy et al., 2020; Hartvigsen et al., 2022; Lee et al., 2022). To reduce such bias, implicit hate speech, toxic language use without any explicit hateful words or phrases, has been explored (Kennedy et al., 2018; ElSherief et al., 2021; Hartvigsen et al., 2022).

**Beyond Binary Toxicity Detection**  Treating toxicity detection as a binary task may not be enough to understand nuanced intents and people's reactions to toxic language use (Jurgens et al., 2019; Rossini, 2022). To holistically analyze toxicity, recent works take a more fine-grained and multidimensional approach: (1) **Explainability** explains why a particular chat is toxic with highlighted rationales (Mathew et al., 2021), free-text annotations of implied stereotype (Sap et al., 2020; ElSherief et al., 2021; Sridhar and Yang, 2022), or pre-defined violation norms (Chandrasekharan et al., 2018; Park et al., 2021). These explanations can be used not only to improve the performance of the toxicity detection model, but also to train models that generate explanations; (2) **Target identification** finds the targets of toxic speech, such as whether the target is an individual or a group, or the name of the group (e.g., race, religion, gender) (Ousidhoum et al., 2019; Mathew et al., 2021); (3) **Context sensitivity** determines toxicity by leveraging context, such as previous tweets (Menini et al., 2021), comments (Pavlopoulos et al., 2020; Xenos et al., 2021) or previous sentences and phrases within the comments (Gong et al., 2021). They show that context can alter labeling decisions by annotators, but that it does not largely impact model performance (Pavlopoulos et al., 2020; Xenos et al., 2021; Menini et al., 2021); (4) **implication** understands veiled toxicity that are implied in codewords and emojis (Taylor et al., 2017; Lees et al., 2021), and

microaggressions that subtly expresses a prejudice attitude toward certain groups (Breitfeller et al., 2019; Han and Tsvetkov, 2020); and (5) **Subjectivity** measures annotation bias (Sap et al., 2022) and manage annotator subjectivity involved in labeling various types of toxicity, which arises from differences in social and cultural backgrounds (Davani et al., 2022). In this paper, we analyze the toxicity of *synchronous* conversations in terms of the aforementioned dimensions by identifying explanation of toxicity as a form of norm categories (*explainability*), finding targets of HIB words (*target identification*), leveraging context for both annotation and modeling (*context sensitivity*), asking annotators for implied knowledge statement (*implication*), and examining how human decisions align with machine decisions under different amounts of information (*subjectivity*).

## 5   Conclusion

In this paper, we analyzed messages flagged by human moderators on Twitch to understand the nature of norm violations in live-stream chats, a previously overlooked domain. We annotated 4,583 moderated chats from live streams with their norm violation category and contrasted them with those from asynchronous platforms. We shed light on the unique characteristics of live-stream chats and showed that models trained with existing data sets perform poorly in detecting toxic messages in our data, which motivates the development of specialized approaches for the synchronous setting. Our experiments established that selecting relevant context is an important feature for detecting norm violations in the synchronous domain.  we hope our work will help develop tools that enable human moderators to efficiently moderate problematic comments in real-time synchronous settings and make the user-experience in these communities more pleasant.

## 6  Limitations

Our data, analysis, and findings have certain limitations. Our research is restricted to the English language and the Twitch platform, although the methods used to detect rule violations in live-stream chat and collect data can be adapted to other languages. Additionally, we recognize that our annotators were recruited from one country, which may result in a lack of diversity in perspectives and potential societal biases. Furthermore, we established a 2-minute context window for each moderated comment within the moderation event, but this may not capture all relevant context.

Additionally, the small size of our human-annotated data may limit the generalizability of our findings to other situations. We recognize that our data set may not represent all instances of rule violations in real-world scenarios. This may be due to the biases of the moderators in choosing which users or comments to moderate or prioritizing certain types of violations over others. Also, the randomly sampled data we annotated may not be representative of the entire population and the imbalance of rule violation classes in our data set may not contain enough samples of rare categories to make definitive conclusions.

Our experimental results indicate that models trained to detect norm violation using our data are far from perfect and may produce errors. When such models are used in real world applications, this can result in overlooking potentially problematic comments or incorrectly flagging non-problematic comments. Therefore, we recommend using AI-based tools to assist human moderators rather than trying to fully replace them. Practitioners should also be aware that there may be users with malicious intent who try to bypass moderation by making their comments appear innocent. By employing moderation models, malicious users may be better able to craft toxic messages undetectable by existing models. As mentioned above, having a final step of human review or verification of the model output will be beneficial. Additionally, it may be necessary to continuously update the model and limit public access to it.

## 7  Ethical Considerations

We took several steps to ensure that our data collection was ethical and legal. We set the hourly rate of compensation for workers at $16.15, which was well above the country's minimum wage at the time ($7.4). To ensure the safety and well-being of our workers, we maintained open communication channels, allowing them to voice any question, concerns, or feedback about the data annotation. This also helped to improve the quality of the collected data as we promptly addressed issues reported by workers throughout the process. We also give each annotation instance enough time so that we do not pressure annotators (40 days for 4,583 instances). We did not collect any personal information from annotators and we did not conduct any experiments with human subjects.

We confirm that we collected and used chats, also referred to as user content, in accordance with Twitch's Terms of Service and do not publicly release the data as it may be in violation of laws against unauthorized distribution of user content. However, we intend to make the platform-specific knowledge statements we compiled available to support future research on real-time chat in the live-streaming domain. During the collection process, we used the official Twitch API to monitor and retrieve chats.

Lastly, we want to emphasize that careful consideration must be given to user privacy when using moderation events to study norm violations. While users may be aware that their comments can be viewed by others in the chat room, researchers must also understand that users have the right to request not to be included in the data and establish a mechanism for users to contact researchers to have their data removed, and refrain from publicly releasing the data and instead share it on a need-to-know basis to control who has access to the data.

## 8  Acknowledgement

We would like to thank to Yeeun Shin (SoftlyAI) for managing the data annotation process and Kyumin Park (SoftlyAI) for the development of the web annotation framework. Datumo, known as SELECTSTAR in South Korea, provided a crowd-sourcing platform for the annotation of the data.

This work was funded in part by the Defense Advanced Research Projects Agency (DARPA) under Contract No. N660011924033, Contract No. HR00112290106, and Contract No. HR00112290025, and with support from the Keston Exploratory Research Award. The views and conclusions contained herein are those of the authors and should not be interpreted as necessarily representing the official policies, either expressed or implied, of DARPA, ARO or the U.S. Government.

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

# A Annotation Details

We engage in active discussions with annotators and provide detailed feedback after multiple rounds of pilot study to ensure the data quality.

## A.1 Annotation UI

To make it easy for annotators to annotate with various types of contexts, we create an annotation tool. The annotation tool has three options and the user can select each option for each step annotation. Figure 6 shows the UI for step 1 which shows only the user's last chat (bad utterance) before the moderation event. Figure 7 shows chat logs up to two minutes ago based on the moderation events on *multi user context* panel. To make it easier for annotators to find previous chats from moderated users, we create *single user context* panel to only display chat logs of the moderated user in multi user context. Figure 8 shows both chat logs and video context. The video context shows 1-minute clipped video around the moderation event.

## A.2 Annotation Consolidation

To determine the final label for each moderated event, we aggregate the labels of annotators using a majority vote with heuristic rules. Each annotator $a_i$ identifies a list of violated rules $\mathcal{L} = \{l_1, l_2, \cdots, l_k\}$ for a moderated event $e$ at each stage $k = \{1, 2, 3\}$. Here, we don't consider the target for HIB. We first evaluate the percentage agreement to measure inter-annotator agreement in each stage by *exact match* and *partial match*. The *exact match* determines whether all three annotators have exactly the same rules ($\mathcal{L}_{a1} = \mathcal{L}_{a2} = \mathcal{L}_{a3}$) and *partial match* determines whether there is at least one intersection rule between three annotators ($(\mathcal{L}_{a1} \cap \mathcal{L}_{a2} \cap \mathcal{L}_{a3}) > 0$). Table 4 shows the inter-annotator agreement percentage. We find that 98% of agreements from *exact match* are single label cases (i.e., 98% of exact matches have only one label) and many disagreements are resolved using the *partial match* method. 92% disagreements that persist even with the *partial match* method are the case where one or two annotators marking a comment as violating the "Incivility" rule while the others do not. Finally, to determine the gold label using the annotations from the three annotators, we apply a *majority vote* approach, choosing the rule types that were selected by at least two people. We discard approximately 3% of events that cannot be consolidated because all three annotators provided different labels.

| Majority Vote | Stage 1 | Stage 2 | Stage 3 |
|---|---|---|---|
| Non-identifiable | 57.55% (545) | 1.35% (16) | 0.24% (3) |
| Broadcaster | 6.55% (62) | 59.83% (712) | 59.71% (744) |
| OIB | 7.07% (67) | 26.89% (320) | 32.34% (403) |
| OOB | 0.32% (3) | 1.09% (13) | 1.61% (20) |
| Disagreement | 28.51% (270) | 10.84% (129) | 6.10% (76) |
| Total | 947 | 1,190 | 1,246 |

Table 8: **Inter-annotator Percent Agreement for Targets of HIB**. Presents the agreement percentage of HIB after majority vote. The numbers in parentheses indicate the absolute number of events.

| Category | Platform | Streamer | Total |
|---|---|---|---|
| Discrimination | 2 | 2 | 4 |
| HIB | 73 | 60 | 133 |
| Privacy | 0 | 0 | 0 |
| Inappropriate Contents | 0 | 0 | 0 |
| Off Topic | 1 | 0 | 1 |
| Spam | 25 | 25 | 50 |
| Meta-Rules | 17 | 25 | 42 |
| Incivility | 69 | 71 | 140 |
| Total | 187 | 183 | 370 |

Table 9: **Data statistics of knowledge statements**.

**Target Agreement for HIB** For cases consolidated as HIB with the majority vote, we further analyze the inter-annotator agreement of *target* labels among annotators who have marked them as HIB. In cases where the annotator was unable to identify the target, we asked them to mark the target as "non-identifiable". Table 8 shows that most HIB words (92.05%) are directed at someone in the broadcast such as the broadcaster or viewers.

## A.3 Knowledge Statement.

**Template.** Table 2 shows templates for the knowledge statement. For platform knowledge, annotators should fill out the span and what span means, and choose whether the span is emoji or text. For streamer knowledge, annotators should fill out the name of streamer and his/her personal background that may need to decide the label.

**Data Statistics.** Table 9 shows data statistics of knowledge statement for each coarse-level norm category. Statistics show that 296 examples (6.6%) require knowledge, with 183 examples requiring streamer knowledge and 187 examples requiring platform knowledge. Note that there are some examples require both. Statistics demonstrate that HIB and incivility require domain knowledge the most to understand the meaning behind them.

**Knowledge statement examples.** For each coarse-level norm category, we present its examples (See Table 10).

| Category | Knowledge | Knowledge Statements |
|---|---|---|
| Discrimination | Platform | [emote] is emoji that means Turkish. |
| | Streamer | [streamer] is playing a Chinese game. |
| HIB | Platform | ResidentSleeper is emoji that means boredom, originating from a man who fell asleep on stream. |
| | Streamer | [streamer] was LoL(League of Legends) game streamer, but he seems to quit and play gamble. |
| Spam | Platform | Gamba is text that means gambling and the [streamer] ordered mods to ban whoever type it. |
| | Streamer | [streamer] has banned using emoji "PogU". |
| Meta-Rules | Platform | Shoutout is text that means highlight notable members in chat, prompting others to follow their channel. |
| | Streamer | [streamer] is a game streamer on Twitch. |
| Incivility | Platform | !sac or !sacme is text that means sacrifice. People in the chat sacrifice themselves(get timed out) by typing them to earn some kind of points. |
| | Streamer | [streamer] has declined [person]'s fight offer. |
| Off-topic | Platform | Tankies is emoji that means implying supports toward Russia (or Soviet Union). |

Table 10: **Knowledge statement examples.**

## A.4 Examples of Incivility.

Table 11 presents examples of chat moderation by streamers where the underlying reason for moderation is not apparent. The cases highlight potentially uncomfortable situations that streamers may encounter.

| Chat | Action |
|---|---|
| I'm 11 so I'm really sad | Ban |
| My mum said she wants to marry you | Ban |

Table 11: **Example cases of Incivility.**

## B Experimental Setup Details

Each fine-tuned experiment uses 1 NVIDIA RTX A5000 GPU and uses FP16. We implement models using PyTorch (Paszke et al., 2019) and Huggingface Transformers (Wolf et al., 2019). We use the Adam optimizer with a maximum sequence length of 256 and a batch size of 4. We set 100 epochs and validate the performance every 100 steps. The stopping criteria is set to 10. For each data, we searched for the best learning rate for our model out of [1e-5, 2e-5, 5e-5, 1e-4, 3e-4]. Then, we report the average score of 3 runs by different random seeds (42, 2023, 5555). Each run takes 10 to 30 minutes. To determine the data distribution ratio between positives and negatives in the training data, we searched for the best distribution out of [1:1, 1:2, 1:5, Original] by random negative sampling. As shown in Table 12, we found that the evenly distribution (1:1) shows the most stable performance with the lowest standard deviation under with and without context. Data statistics for both Twitch and Reddit (Park et al., 2021) are presented in Table 13-14. Note that we report the number of data statistics after sampling the same number of negative samples as positive samples.

## C Ablation Study

## C.1 Context Arrangement

To understand how the context arrangement in the input affects the performance, we conduct experiments with multiple variants of context arrangement on moderation detection (See Table 15). First, the results show that randomly shuffled context consistently harm the performance. It indicates that context order matters, in contrast to the findings in dialog system study results (Sankar et al., 2019; He et al., 2021). Moreover, input as the sequential order of chats presented in the context-aware model (Pavlopoulos et al., 2020), or adding more contexts (*e.g.,* broadcast category, rule text) degrade the performance. This indicates that the target text should always be placed first, and some contexts may not be helpful.

| Category | No Context | | | | Multi-user Context (utterance) | | | | Multi-user Context (first) | | | |
|---|---|---|---|---|---|---|---|---|---|---|---|---|
| | 1:1 | 1:2 | 1:5 | Original | 1:1 | 1:2 | 1:5 | Original | 1:1 | 1:2 | 1:5 | Original |
| Discrimination | $0.11_{\pm0.01}$ | $0.20_{\pm0.03}$ | $0.41_{\pm0.01}$ | $0.00_{\pm0.00}$ | $0.04_{\pm0.00}$ | $0.06_{\pm0.02}$ | $0.06_{\pm0.11}$ | $0.00_{\pm0.00}$ | $0.05_{\pm0.01}$ | $0.09_{\pm0.09}$ | $0.12_{\pm0.20}$ | $0.00_{\pm0.00}$ |
| HIB | $0.52_{\pm0.01}$ | $0.52_{\pm0.03}$ | $0.46_{\pm0.01}$ | $0.14_{\pm0.25}$ | $0.61_{\pm0.05}$ | $0.64_{\pm0.02}$ | $0.20_{\pm0.35}$ | $0.00_{\pm0.00}$ | $0.61_{\pm0.01}$ | $0.48_{\pm0.26}$ | $0.40_{\pm0.35}$ | $0.20_{\pm0.36}$ |
| Privacy | $0.05_{\pm0.03}$ | $0.05_{\pm0.03}$ | $0.11_{\pm0.07}$ | $0.00_{\pm0.00}$ | $0.00_{\pm0.00}$ | $0.02_{\pm0.01}$ | $0.08_{\pm0.02}$ | $0.00_{\pm0.00}$ | $0.01_{\pm0.00}$ | $0.07_{\pm0.09}$ | $0.07_{\pm0.02}$ | $0.00_{\pm0.00}$ |
| Inapt. Contents | $0.12_{\pm0.01}$ | $0.58_{\pm0.07}$ | $0.33_{\pm0.06}$ | $0.42_{\pm0.36}$ | $0.09_{\pm0.03}$ | $0.36_{\pm0.10}$ | $0.62_{\pm0.03}$ | $0.36_{\pm0.33}$ | $0.11_{\pm0.02}$ | $0.28_{\pm0.01}$ | $0.64_{\pm0.03}$ | $0.38_{\pm0.34}$ |
| Off Topic | $0.07_{\pm0.00}$ | $0.19_{\pm0.12}$ | $0.28_{\pm0.03}$ | $0.00_{\pm0.00}$ | $0.10_{\pm0.04}$ | $0.13_{\pm0.06}$ | $0.18_{\pm0.16}$ | $0.00_{\pm0.00}$ | $0.08_{\pm0.03}$ | $0.15_{\pm0.08}$ | $0.37_{\pm0.07}$ | $0.00_{\pm0.00}$ |
| Spam | $0.63_{\pm0.01}$ | $0.68_{\pm0.00}$ | $0.67_{\pm0.04}$ | $0.64_{\pm0.04}$ | $0.66_{\pm0.01}$ | $0.72_{\pm0.01}$ | $0.71_{\pm0.03}$ | $0.69_{\pm0.04}$ | $0.70_{\pm0.03}$ | $0.74_{\pm0.01}$ | $0.73_{\pm0.03}$ | $0.75_{\pm0.04}$ |
| Meta-Rules | $0.65_{\pm0.01}$ | $0.71_{\pm0.02}$ | $0.48_{\pm0.42}$ | $0.69_{\pm0.04}$ | $0.65_{\pm0.04}$ | $0.74_{\pm0.01}$ | $0.00_{\pm0.00}$ | $0.00_{\pm0.00}$ | $0.62_{\pm0.02}$ | $0.68_{\pm0.04}$ | $0.00_{\pm0.00}$ | $0.24_{\pm0.42}$ |
| Incivility | $0.28_{\pm0.04}$ | $0.09_{\pm0.15}$ | $0.09_{\pm0.16}$ | $0.00_{\pm0.00}$ | $0.24_{\pm0.12}$ | $0.31_{\pm0.20}$ | $0.08_{\pm0.14}$ | $0.00_{\pm0.00}$ | $0.45_{\pm0.03}$ | $0.46_{\pm0.04}$ | $0.00_{\pm0.00}$ | $0.00_{\pm0.00}$ |

Table 12: **Experimental Results on different Training data distribution.** macro F1 score for each coarse-level norm category, and scores are average of 3 runs (3 random seeds). Excluding models with an F1 score of 0, the model with the lowest standard deviation is **bold** for each category and its context setting.

| Category | Stage 1 | | | | | | Stage 2 | | | | | | Stage 3 | | | | | |
|---|---|---|---|---|---|---|---|---|---|---|---|---|---|---|---|---|---|---|
| | Train | | Development | | Test | | Train | | Development | | Test | | Train | | Development | | Test | |
| | 1 | 0 | 1 | 0 | 1 | 0 | 1 | 0 | 1 | 0 | 1 | 0 | 1 | 0 | 1 | 0 | 1 | 0 |
| Discrimination | 83 | 83 | 10 | 856 | 9 | 863 | 81 | 81 | 9 | 857 | 9 | 863 | 84 | 84 | 9 | 857 | 10 | 862 |
| HIB | 752 | 752 | 106 | 760 | 88 | 784 | 949 | 949 | 121 | 745 | 118 | 754 | 996 | 996 | 124 | 742 | 124 | 748 |
| Privacy | 11 | 11 | 1 | 865 | 1 | 871 | 12 | 12 | 1 | 865 | 1 | 871 | 12 | 12 | 1 | 865 | 1 | 871 |
| Inapt. Contents | 47 | 47 | 3 | 863 | 4 | 868 | 42 | 42 | 3 | 863 | 4 | 868 | 42 | 42 | 3 | 863 | 4 | 868 |
| Off Topic | 70 | 70 | 8 | 858 | 8 | 864 | 78 | 78 | 8 | 858 | 9 | 863 | 77 | 77 | 8 | 858 | 9 | 863 |
| Spam | 789 | 789 | 92 | 774 | 103 | 769 | 935 | 935 | 114 | 752 | 114 | 758 | 926 | 926 | 114 | 752 | 114 | 758 |
| Meta-Rules | 183 | 183 | 21 | 845 | 23 | 849 | 762 | 762 | 91 | 774 | 94 | 778 | 762 | 762 | 92 | 774 | 94 | 778 |
| Incivility | 1,607 | 1,607 | 191 | 675 | 198 | 674 | 718 | 718 | 88 | 778 | 91 | 781 | 678 | 678 | 84 | 782 | 84 | 788 |
| ALL | 3,542 | 3,499 | 432 | 434 | 434 | 438 | 3,577 | 3,464 | 435 | 431 | 440 | 432 | 3,577 | 3,464 | 453 | 431 | 440 | 432 |

Table 13: **Train/Dev/Test Statistics of Normvio-RT.**

| Category | Train | | Development | | Test | |
|---|---|---|---|---|---|---|
| | 1 | 0 | 1 | 0 | 1 | 0 |
| Incivility | 1,787 | 1,787 | 252 | 4,962 | 230 | 4,901 |
| Harassment | 5,048 | 5,048 | 605 | 4,609 | 546 | 4,585 |
| Spam | 3,649 | 3,649 | 418 | 4,796 | 417 | 4,714 |
| Off Topic | 3,009 | 3009 | 326 | 4888 | 331 | 4800 |
| Hate Speech | 4,930 | 4,930 | 607 | 4,607 | 667 | 4,464 |
| Content | 20,614 | 20,614 | 2,773 | 2,441 | 2,618 | 2,513 |

Table 14: **Train/Dev/Test Statistics of Normvio (Park et al., 2021).**

| Arrangement | Context | F1 Score |
|---|---|---|
| text | - | 0.703 |
| text + [SEP] + context | single_user | 0.747 |
| | multi_user (event) | **0.701** |
| | multi_user (utterance) | **0.908** |
| | multi_user (first) | **0.955** |
| text + [SEP] + $RAND$(context) | single_user | 0.708 |
| | multi_user (event) | 0.671 |
| | multi_user (utterance) | 0.881 |
| | multi_user (first) | 0.952 |
| context + [SEP] + text (Pavlopoulos et al., 2020) | single_user | 0.767 |
| | multi_user (event) | 0.671 |
| | multi_user (utterance) | 0.867 |
| | multi_user (first) | 0.951 |
| text + [SEP] + context + [SEP] + broadcast cat. | single_user | 0.777 |
| | multi_user (event) | 0.671 |
| | multi_user (utterance) | 0.904 |
| | multi_user (first) | 0.941 |
| text + [SEP] + context + [SEP] + rule text | single_user | **0.781** |
| | multi_user (event) | 0.671 |
| | multi_user (utterance) | 0.895 |
| | multi_user (first) | 0.953 |

Table 15: **Performance on moderation detection by different context arrangement.** macro F1 score for "All" in Table 6. Best models for each context are **bold**.

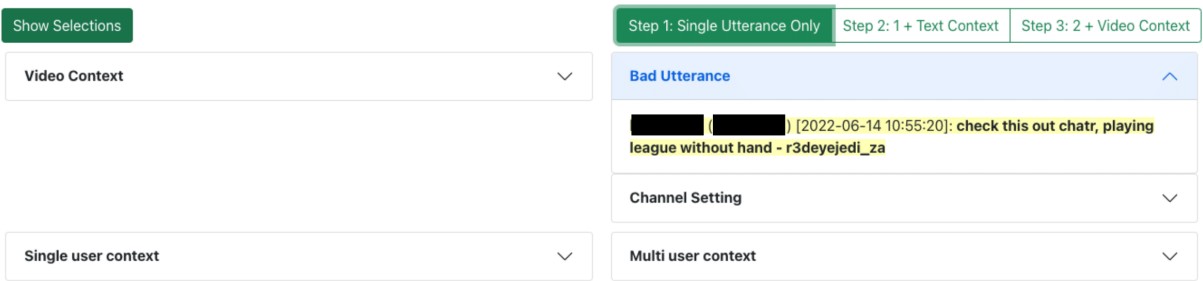

Figure 6: **Step 1. single utterance** shows only the user's last chat before the moderation event.

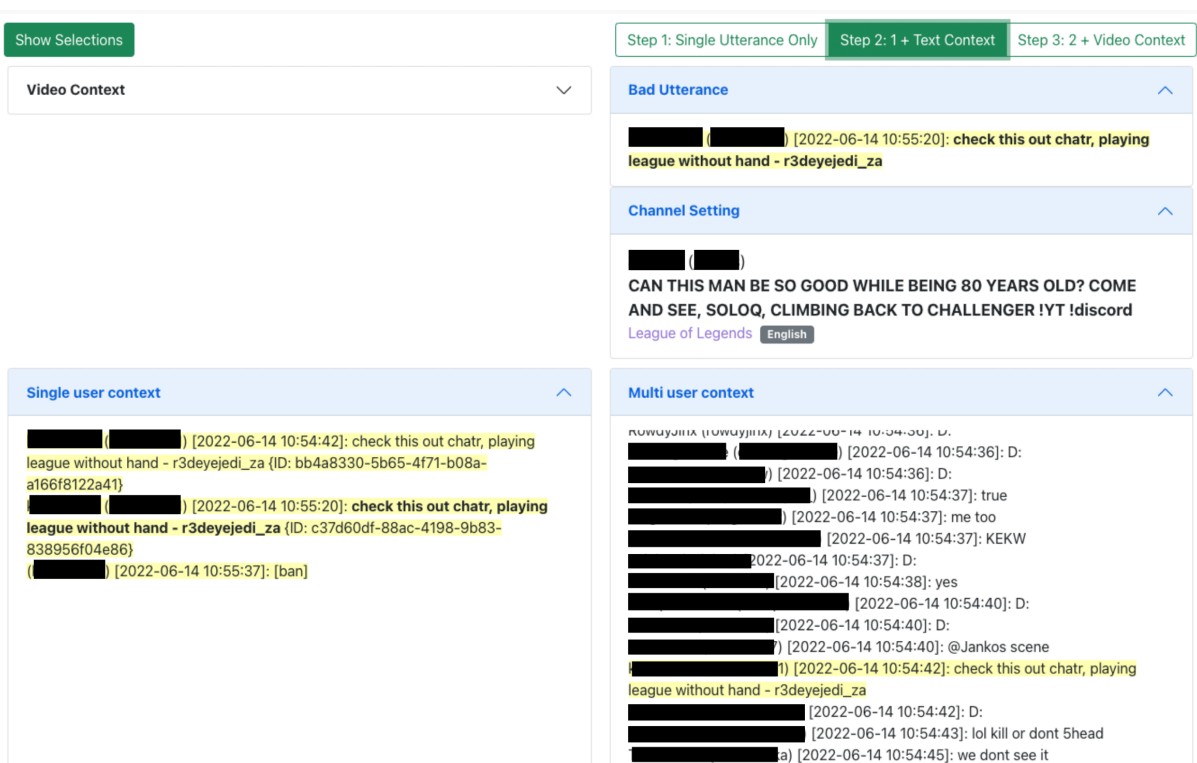

Figure 7: **Step 2. + chat context** shows chat logs up to two minutes ago based on the moderation events (multi user context). single user context only shows the moderated user's messages within two minutes.

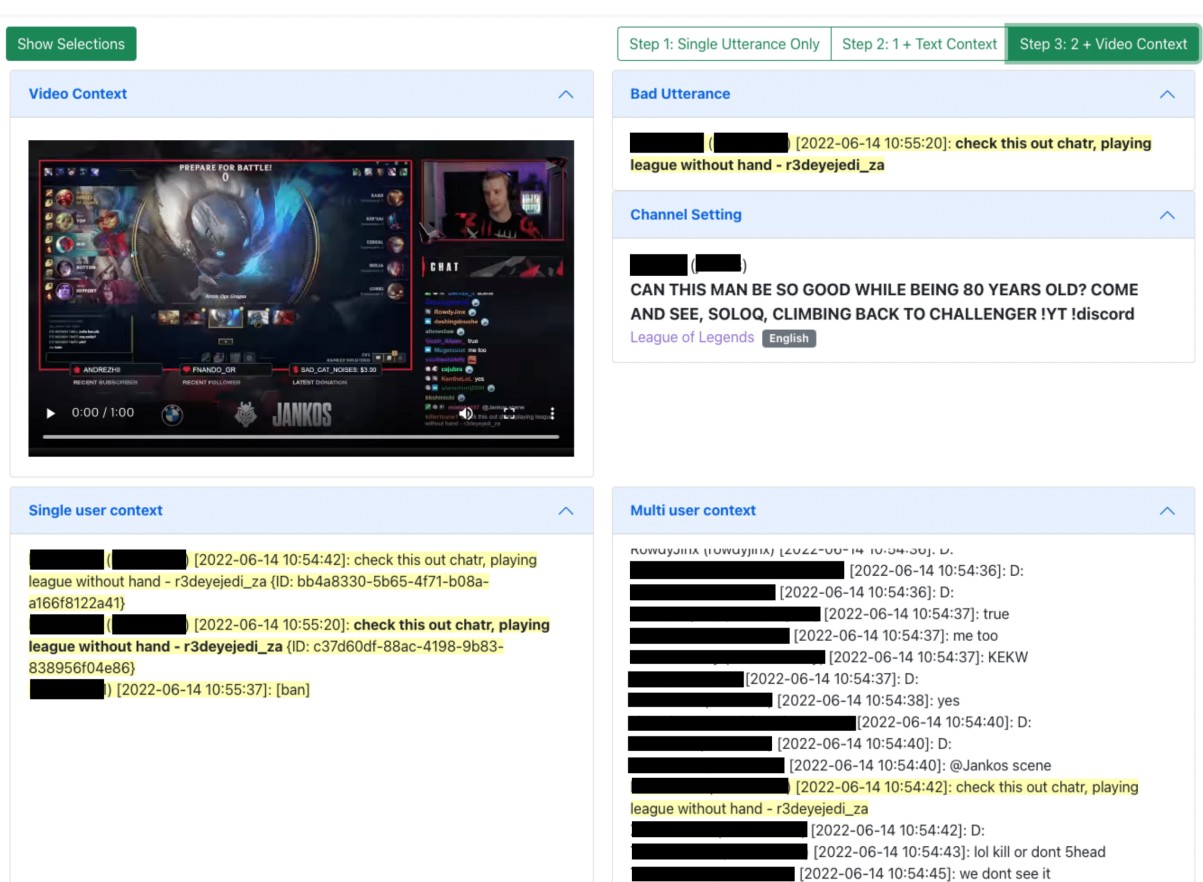

Figure 8: **Step 3. + video context** shows both chat logs and video context.