# OpenReview forum: "Analyzing Norm Violations in Live-Stream Chat"
_EMNLP/2023/Conference — EMNLP 2023 Main_

### Official Review · Reviewer_9tw6 · 2023-08-05

**Paper Topic And Main Contributions:** 1. The paper proposes a new task
**Soundness:** 4

**Excitement:**

4: Strong: This paper deepens the understanding of some phenomenon or lowers the barriers to an existing research direction.

**Questions For The Authors:**

1. Isn't the claim that the additional context helps in determining norm-violations fairly evident? why is this interesting?
2. The same question is true for how additional context helps with inter-annotator agreement.

**Reasons To Accept:**

1. The topics of the paper is relevant for the NLP community: detecting norm violations in live-steam media conversations. The previous work in this space has only been on non-live services such as Reddit
2. The methodology is thorough with all relevant details that support the claims
4. The paper is well presented with clear details on data collection, annotation strategy, experimentation, and analysis
5. The collected and norm-categorized data might be useful for further research in this direction

**Reasons To Reject:**

1. The authors mention that the existing models have bad precision for toxicity detection but don’t explain why. Further analysis on this end would be helpful
2. More information is needed for the experimental setup

**Reproducibility:**

4: Could mostly reproduce the results, but there may be some variation because of sample variance or minor variations in their interpretation of the protocol or method.

**Reviewer Confidence:**

4: Quite sure. I tried to check the important points carefully. It's unlikely, though conceivable, that I missed something that should affect my ratings.

---

> ### Author Rebuttal · Authors · 2023-08-25
>
> We appreciate your feedback on our work. Thank you for taking the time to provide us with valuable insights. Here are our comments on your questions.
>
> #### **R1. Rationale behind bad precision of existing toxicity detection models**
>
> Our findings suggest that the language used in online communities with a bulletin board format can differ from typical chat conversations, resulting in a shift in data distribution and synchronous/asynchronous communication styles. Additionally, the Twitch-trained model demonstrates a high level of generalizability (Table 7), indicating that typical chats may encompass a broader range of toxic behavior.
> As a result, we believe that existing toxicity detection models trained on internet data from platforms like Twitter and Reddit may be adversely affected by this disparity.
>
> #### **R2. Details of experimental setup**
>
> Apologies for any misunderstanding. We will strengthen our experimental setup in more detail.
>
> #### **Q1. The assertion that additional context is helpful for detection norm-violation and inter-annotator agreement appears to be self-evident.**
>
> I agree that it is evident that additional context can be helpful to detecting norm violation. Moreover, it can be helpful to any other NLP tasks as well (e.g., open-book QA).
> However, we demonstrate the importance of determining which specific context for detecting norm violation in a live-streaming environment which is beyond simple incorporation of context.

---

### Official Review · Reviewer_YGch · 2023-08-05

**Soundness:** 4

**Excitement:**

4: Strong: This paper deepens the understanding of some phenomenon or lowers the barriers to an existing research direction.

**Paper Topic And Main Contributions:**

This paper analyzes norm violations and toxic language in live stream chats, which have different characteristics compared to asynchronous platforms like Reddit or Twitter that have been the focus of most prior work. The authors collected a new dataset called NormVio-RT containing 4,583 moderated Twitch comments annotated for 15 fine-grained norm violation types. Experiments find existing toxicity detection models perform poorly on live stream data, motivating new approaches. Adding relevant context improves inter-annotator agreement on labels. Models using context outperform those without, especially for norms relying on relationships between messages.

**Questions For The Authors:**

A few small questions:
- In 3.2, how to apply the (6) Rule text-based model to testing data, considering the violated rule is unknown at prediction time?
- I don't understand this sentence:"The intuition for this selection is that the modera-391 tion event may have taken place much earlier than 392 the moderation event." What does it mean by moderation event is much earlier than moderation event?
- An specific example showing why (4) Multi-user context is useful would be great for readers to understand the paper.
- I am very curious about the performance of the proposed model in other domains, such as YouTube live comments.

**Reasons To Accept:**

- The proposed research problem is very interesting and has great practical impact.
- It shows several interesting insights on this problem.

**Reasons To Reject:**

none.

**Reproducibility:**

4: Could mostly reproduce the results, but there may be some variation because of sample variance or minor variations in their interpretation of the protocol or method.

**Reviewer Confidence:**

3: Pretty sure, but there's a chance I missed something. Although I have a good feel for this area in general, I did not carefully check the paper's details, e.g., the math, experimental design, or novelty.

---

> ### Author Rebuttal · Authors · 2023-08-25
>
> We appreciate your feedback on our work. Thank you for taking the time to provide us with valuable insights. Here are our comments on your questions.
>
> #### **Q1. `Rule Text` experiments**
>
> As mentioned in line 400, we do not use rule text for the inference since we do not know which rule is violated for unseen examples. We make an inference based on chat as it is without any context. This experimental procedure aims to assess whether the model possesses an intrinsic understanding of the underlying rationale behind norm violations.
>
> #### **Q2. Meaning of `The intuition for this selection is that the moderation event may have taken place much earlier than the moderation event`**
>
> Apologies for any misunderstanding. The broadcaster or moderator can take appropriate moderation action within a few minutes of encountering problematic behavior. It is important to note that this action may not be instantaneous, resulting in a temporal gap between the user’s problematic behavior and the moderation action taken by the broadcaster or moderator.
>
> #### **Q3. Examples of scenarios in which the use of a `multi-user context` can contribute to improvement**
>
> Thank you for your suggestion. We acknowledge the value of providing real-world examples as they can be helpful in illustrating our points. However, incorporating actual instances into the paper proved challenging. Nevertheless, we would like to provide the following example cases:
> ```
> Problematic chat: 'GO M4CAC0S'
> ```
> The problematic chat can be challenging to identify as a violation of social norms for individuals who are unfamiliar with internet terminology. However, within a multi-user context, it was noticed that all the conversations taking place were constantly focused on making discriminatory statements towards Asian gamers. This surrounding context provides a clearer understanding of why this specific chat can be considered an example of discrimination.
>
> #### **Q4. Performance of model in other domains (YouTube live comments)**
>
> We appreciate your suggestion and are eager to expand our efforts in developing a comprehensive framework for detecting toxic content in all types of synchronous chat platforms.

---

### Official Review · Reviewer_HcKC · 2023-08-07

**Soundness:** 5

**Excitement:**

4: Strong: This paper deepens the understanding of some phenomenon or lowers the barriers to an existing research direction.

**Missing References:**

https://dl.acm.org/doi/abs/10.1145/3345645.3351104
https://aclanthology.org/2022.wnut-1.7/

**Paper Topic And Main Contributions:**

Authors analyze synchronous and asyc chat message discrepancy and the important role of context in sync chats. The authors conducted analysis on Twitch stream chats and use Twitch channel moderated events and rules to build a benchmark dataset. The key highlight is that utilizing context helps improve the inter-annotator agreement and also the model performance.

**Reasons To Accept:**

Very well written paper and I would say the first to analyze synchronous and asyc chat message discrepancy and the important role of context in sync chats
Good problem framing
Comparison against async chats e.g. on Reddit
Annotated dataset


**Reasons To Reject:**

No reason

**Reproducibility:**

4: Could mostly reproduce the results, but there may be some variation because of sample variance or minor variations in their interpretation of the protocol or method.

**Reviewer Confidence:**

5: Positive that my evaluation is correct. I read the paper very carefully and I am very familiar with related work.

---

> ### Author Rebuttal · Authors · 2023-08-25
>
> We appreciate your positive feedback on our work. Rest assured, we will add the references you have recommended into our content. Thank you for taking the time to provide us with valuable insights.

---

### Meta-Review · Area_Chair_YWzz · 2023-09-21

**Recommendation:** 5

**Metareview:**

This paper presents a study on norm violations in live-streaming platforms. It introduces a new data set of moderated comments from Twitch and a taxonomy of 15 norm violation types. Models are trained that use context to identify norm violations that can improve overall moderation performance.

The reviewers are in agreement in their positive judgement of the paper. Overall, they consider the paper is well writen and appreciate the problem framing and importance of the work. The reviewers also appreciated the novelty of the work in dealing with live / synchroneuous chat and the insights gained into this problem, especially comparing it with async platform.

The data and annotation process are clear and it should form a solid basis for future work in this space.

The reviewers did not identify any substantial reason for rejection or area of improvement.

---

### Decision · Program_Chairs · 2023-10-07

**Decision:**

Accept-Main

**Comment:**

This paper presents a study on norm violations in live-streaming platforms. It introduces a new data set of moderated comments from Twitch and a taxonomy of 15 norm violation types. Models are trained that use context to identify norm violations that can improve overall moderation performance.

The reviewers are in agreement in their positive judgement of the paper. Overall, they consider the paper is well writen and appreciate the problem framing and importance of the work. The reviewers also appreciated the novelty of the work in dealing with live / synchroneuous chat and the insights gained into this problem, especially comparing it with async platform.

The data and annotation process are clear and it should form a solid basis for future work in this space.

The reviewers did not identify any substantial reason for rejection or area of improvement.